# COVID-19 Vaccine Booster Hesitancy among Hispanic Adults: A Cross-Sectional Study from the Puerto Rico Community Engagement Alliance against COVID-19 Disparities (PR-CEAL)

**DOI:** 10.3390/vaccines11091426

**Published:** 2023-08-28

**Authors:** Hérmilis Berríos, Andrea López-Cepero, Cynthia M. Pérez, Stephanie Cameron, Adriana D. Pons Calvo, Vivian Colón-López

**Affiliations:** 1Division of Cancer Control and Population Sciences, University of Puerto Rico Comprehensive Cancer Center, San Juan 00927, Puerto Rico; hermilis.berrios@upr.edu (H.B.); stephanie.cameron@upr.edu (S.C.); adriana.pons@upr.edu (A.D.P.C.); 2Department of Epidemiology, Rollins School of Public Health, Emory University, Atlanta, GA 30322, USA; andrea.lopez11@upr.edu; 3Department of Biostatistics and Epidemiology, University of Puerto Rico Medical Sciences Campus, San Juan 00935, Puerto Rico; cynthia.perez1@upr.edu

**Keywords:** COVID-19, booster, vaccine uptake

## Abstract

Hispanic/Latino communities have suffered a disproportionate burden of the COVID-19 pandemic. Although Puerto Rico has one of the highest COVID-19 primary series vaccination rates nationwide, this estimate contrasts with the reported booster doses’ low uptake. This study aimed to assess health belief correlates of COVID-19 vaccine booster uptake. Using a convenience sampling approach, the Puerto Rico-Community Engagement Alliance (PR-CEAL) conducted a cross-sectional study where 787 participants were recruited using online and in-person strategies between December 2021 and February 2022. Participants were adults 18 years or older, Spanish-speaking, and residents of Puerto Rico. The Health Belief Model was used to evaluate attitudes and beliefs. A total of 784 participants were used in this analysis. Adjusted Poisson regression models were used to estimate prevalence ratios (PR) and 95% confidence intervals (CI) of booster refusal. Overall, 22% of participants refused the vaccine booster or had not gotten it yet. Adjusted models showed that (i) participants who disagreed that getting the booster dose either made them feel less worried about COVID-19 or (ii) felt that the vaccine decreased their chances of getting COVID-19 presented higher booster-refusal prevalence ratios (PR = 4.20, 95% CI: 3.00, 5.90; PR = 3.70, 95% CI: 2.64, 5.18). Moreover, participants that (iii) reported having concerns for booster side effects [PR = 2.47; 95% CI = 1.73, 3.51], (iv) booster efficacy [PR = 2.50; 95% CI = 1.75, 3.58], and (v) booster safety [PR = 2.80; 95% CI = 1.96, 3.99] were significantly more likely to refuse the booster. In conclusion, booster vaccination refusal was associated with lower perceived vaccine benefits and greater barriers among adults in Puerto Rico. These results informed the development of PR-CEAL’s targeted community outreach strategies and public health campaigns to increase booster vaccine uptake.

## 1. Introduction

As of 5 July 2023, the Puerto Rico Department of Health has reported 485,199 confirmed COVID-19 cases and 6106 related deaths [1]. Nine months after the promotion of education strategies for the mitigation and prevention of the spread of the virus, vaccines became available in Puerto Rico for a select group of the population; the remainder of the population gained access to the vaccines in early 2021. The primary series vaccination rate of the COVID-19 vaccine in Puerto Rico was approximately 86.3% [2], one of the highest in the United States (US) [3,4]. However, this extraordinary accomplishment was not sustained in terms of booster completion.

Several studies have discovered that booster doses have not been accepted to the same degree as the primary series of the vaccine [5,6,7]. On October 2021, the Puerto Rico Department of Health (PRDH) reported that booster doses were approved for administration on the island. Booster doses were first administered among people 65 years and older and any person at high risk of getting COVID-19, such as those immunocompromised [8]. Many vaccination centers were established, and several educational campaigns were organized to increase awareness and vaccination intent against COVID-19 [9]. Despite these island-wide efforts, according to the PRDH, only 34.6% of the population has received the booster dose recommended by the Centers for Disease Control and Prevention (CDC) [1,10]. 

Racial/ethnic disparities in vaccine booster uptake have been documented. For the Hispanic population, compared with non-Hispanic Whites, high uptake of the primary series and a remarkably low booster uptake have been reported [11]. It is of primary concern that Hispanics have lower rates of booster uptake compared to non-Hispanic Whites, despite the former disproportionate burden of COVID-19 impact throughout the pandemic compared to the latter [11]. The study by Baker and colleagues documented that Hispanics had higher rates of deaths caused by COVID-19 than non-Hispanic Whites [11]. 

In addition to social and contextual factors influencing booster vaccine uptake, the population’s beliefs and attitudes may also influence booster vaccination. Multiple studies have documented the unwillingness of some groups to receive both the COVID-19 vaccine primary series and booster dose [12,13]. Qin and colleagues reported that the primary reasons for refusing the booster vaccine were the complexity of the vaccination process (e.g., long waiting lines and difficulties in obtaining appointments) and uncertainty about the safety and effectiveness of the vaccine [13]. Recent studies have employed the Health Belief Model (HBM) to understand people’s perceptions and attitudes toward COVID-19 vaccination [12,13,14,15]. The HBM has been widely used to understand health behavior and comprises a series of constructs: perceived susceptibility, perceived severity, perceived benefits, perceived barriers, and cues to action [15]. Quin et al. found that individuals with low perceived susceptibility had lower odds of accepting the booster dose [13]. Additionally, a study evaluating the COVID-19 vaccination intent of the primary vaccine series in Puerto Rico showed that lower perceived susceptibility and vaccination benefits, greater vaccination barriers, and specific cues to action were associated with the unwillingness of adults to receive the COVID-19 vaccine [12]. In the present study, we aimed to understand the predictors of COVID-19 booster vaccine hesitancy using the HBM among adults residing in Puerto Rico. 

## 2. Materials and Methods

### 2.1. Puerto Rico-Community Engagement Alliance (PR-CEAL)

On 16 September 2020, The National Institute of Health (NIH) established the Community Engagement Alliance (CEAL) Consortium to control the pandemic and enhance education about COVID-19. Puerto Rico CEAL (PR-CEAL) was founded in February 2021 as part of the second cohort of consortiums. PR-CEAL has led efforts to address COVID-19 health disparities, misinformation, and mistrust of the vaccination process (including the booster vaccine). Initially, PR-CEAL consisted of five components, including the Community Outreach Engagement Group and four independent research projects. The current analysis uses data from one of the four PR-CEAL research projects. 

### 2.2. Study Design

Along with our nationwide CEAL partners, efforts were made to develop a survey to be implemented across multiple settings. Using this questionnaire (known as the CEAL Common Survey), we conducted a cross-sectional study with a convenience sampling approach. Among the variables included in the survey were sociodemographic characteristics, social determinants of health, COVID-19 prevention behaviors, testing behaviors, trusted sources of information about COVID-19, and general perceptions of COVID-19. The first version of the survey was developed in October 2021. Due to the pandemic’s dynamic nature and since the vaccine’s implementation, a second version of the Common Survey was created in December 2021. This second version included questions about the intent to get a COVID-19 booster vaccine, self-reported chronic conditions, influenza vaccine uptake, and cancer screening practices. Using items developed by Wong et al. [15], PR-CEAL included additional questions based on the HBM to examine COVID-19 vaccination beliefs, attitudes, and behaviors [10,15].

Recruitment started on 30 December 2021, using in-person strategies. Our first recruitment attempt consisted of visits to supermarkets and community settings throughout the island that were previously selected using a probabilistic approach. However, this recruitment effort had to be pivoted after the new Omicron COVID-19 variant was reported in Puerto Rico, yielding only 32 participants. To comply with the latest governmental restrictions implemented to reduce the transmission of COVID-19 [16,17], the team adopted an online recruitment strategy, yielding an additional 755 participants (96% of the total of participants recruited). The team used different platforms, including Facebook, Instagram, and email blasts directed at academic groups, to invite individuals to complete the online questionnaire, which remained open to the public until 8 February 2022. A total of 787 participants were recruited using the two strategies. The study protocol was approved by the University of Puerto Rico Medical Sciences Campus Institutional Review Board (2021, protocol number #6050220). Participants who finished the survey received a $30 gift card sent via certified mail.

### 2.3. Study Population and Study Sample

Study eligibility criteria included persons of 18 years or older, currently residing in Puerto Rico, being able to complete the questionnaire in Spanish, and having access to the internet and an electronic device to complete the survey. Those who were eligible and interested in participating completed the online survey voluntarily. 

### 2.4. Measurements

#### 2.4.1. Main Outcome Variables

Booster acceptance was assessed with the question, “Do you intend to get a booster shot?” as established in the CEAL Common Survey. Response options were “I already got the booster”, “I will get it as soon as possible”, “I am going to wait to see how it affects others”, “I have no intention to get the booster soon, but I could sometime in the future”, and “not at all”. Participants who answered, “I already got the booster”, were classified as ‘accepting the booster vaccine’. Participants who responded to any other category were considered as ‘refusals’. Those participants that answered “I will get it as soon as possible” were categorized as ‘refusal’ considering that the booster vaccine was readily available, and they met the requirements to get the vaccine. 

#### 2.4.2. Main Exposure Variables

Beliefs about the COVID-19 booster vaccine were assessed using the HBM constructs defined by Wong et al. [15]. We examined the following 5 constructs: perceived susceptibility (My chance of getting COVID-19 in the next few months is high; I am worried about the likelihood of getting COVID-19; and Getting COVID-19 is currently a possibility for me), perceived severity (Complications from COVID-19 are serious; I will be very sick if I get COVID-19; and I am afraid of getting COVID-19), perceived benefits (A booster dose is a good idea because it makes me feel less worried about catching COVID-19; and Getting the booster will decrease my chance of getting COVID-19 or its complications), perceived barriers (I worry that the possible side effects of the COVID-19 booster vaccine would interfere with my usual activities; I am concerned about the efficacy of the COVID-19 booster vaccine; and I am worried about the safety of the COVID-19 booster vaccine), and cues to action (I will take the booster only if I am given adequate information about it; and I will take the COVID-19 booster only if the vaccine is taken by many in the public). 

#### 2.4.3. Covariates

Self-reported information on sex, age, educational attainment, ethnicity, annual income, health insurance, and marital status was collected as part of the CEAL Common Survey. These variables have been found to influence people’s COVID-19 thoughts and beliefs [12,15]. Religiosity was assessed with the question, “How important is religiosity for you?”. The response options were ‘not important’, ‘not very important’, ‘somewhat important’, ‘important’, and ‘very important’. Responses were further categorized for analysis as not important, somewhat important, and important/very important due to their distribution in the sample.

### 2.5. Statistical Analysis

Out of the total 787 participants recruited, 784 participants were used for this analysis. We deleted the observations of participants in which information related to the outcome (booster refusal) was unavailable. Associations between sociodemographic characteristics, HBM constructs, and the acceptance of the booster vaccine were evaluated using either a chi-square test or Fisher’s exact test (when appropriate). Prevalence ratios (PR) and 95% confidence intervals (95% CIs) for each HBM construct were computed using Poisson regression models with robust variance errors. The models were adjusted for income, health insurance, marital status, and importance of religion based on bivariate analysis significance. All the statistical analyses were performed using Stata version 17 (StataCorp LLC, College Station, TX, USA). 

## 3. Results

Table 1 shows the participants’ sociodemographic characteristics in the total sample and by booster acceptance status. Most (72.2%) were women, and 71.7% reported having a college degree or greater. Study participants had a mean age of 38.6 (14.1) years. Most participants (96.5%) self-identified as Puerto Rican, and 51.1% indicated that religion was very important to them. Regarding income, 54.5% of participants reported having an annual income lower than $40,000. Regarding COVID-19 booster vaccine outcomes, 611 participants (78%) reported already getting the booster, and 173 participants (22%) refused the booster. A higher income and having health insurance were significantly associated with booster refusal (*p* < 0.05).

Most HBM items were significantly associated with booster refusal, except for the following: “My chance of getting COVID-19 in the next few months is high”, “Getting COVID-19 is currently a possibility for me”, and “I will be very sick if I get COVID-19” (Table 2). Regarding susceptibility, being worried about the likelihood of getting COVID-19 was associated with booster acceptance. For the severity construct, perceiving complications of contracting COVID-19 and being afraid of getting it was associated with booster acceptance. Perceiving benefits for booster vaccination was associated directly with booster acceptance and the cues to action items. Perception of barriers towards the booster dose was associated with booster refusal. 

As shown in Table 3, after adjusting for the covariates, those who were not worried about getting COVID-19 had a higher prevalence (PR = 2.26, 95% CI: 1.61, 3.16) of booster refusal. Those who did not recognize COVID-19 as a severe illness or who reported being afraid of COVID-19 also had a higher prevalence (PR = 3.50, 95% CI: 2.41, 5.07; PR = 2.01, 95% CI: 1.44, 2.80) of booster refusal. In general, the participants who disagreed that getting the booster dose either made them feel less worried about COVID-19 or decreased their chances of getting COVID-19 had a higher prevalence of booster refusal (PR = 4.20, 95% CI: 3.00, 5.90; PR = 3.70, 95% CI: 2.64, 5.18). Similarly, participants who perceived barriers such as being worried about side effects or the efficacy or safety of the booster vaccine had a higher prevalence of booster refusal (PR = 2.47, 95% CI: 1.73, 3.51; PR = 2.50, 95% CI: 1.75, 3.58; PR = 2.80, 95% CI: 1.96, 3.99). 

## 4. Discussion

Our study showed that 22% of the participants in this sample refused the booster vaccine. This estimate is consistent with recent studies documenting high booster-vaccine acceptance or vaccination intent [9,18]. For instance, a study conducted in the US found that approximately 14.1% of the participants were hesitant to receive the booster dose [19]. Another study in China reported that 17.2% of the sample was hesitant about the booster vaccine [13].

The present study findings agree with our prior study on HBM and the intent of the uptake of the initial vaccine series against COVID-19 [12]. This former study showed that participants reported lower perceived vaccine benefits, concerns for booster side effects, booster efficacy, and booster safety were more likely to refuse the booster. Similarly, HBM constructs have been associated with COVID-19 vaccine willingness in other national and international studies. Qin and colleagues (2022) reported that concerns about the safety and effectiveness of the booster were closely related to vaccine hesitancy [13]. Another study by Ghazy et al. reported that one of the most frequent reasons for refusing the booster was the lack of scientific evidence regarding vaccination benefits [20]. This is consistent with our study that showed that participants with low perceived booster vaccination benefits were associated with booster refusal. Similar to our study, Yenew and colleagues also found that greater perceived vaccination barriers were associated with booster refusal [21]. Another study evaluating the intention of parents to vaccinate their children showed that unwillingness to vaccinate was associated with greater parental perception of barriers and lower benefits [22]. As demonstrated in previous studies, HBM constructs are important to understand the intention to vaccinate against COVID-19 [12,15,20,21]. This makes this model paramount to help create campaigns addressing the population’s perceptions and beliefs toward diseases. Our results highlight the need for educating the population to minimize misinformation, barriers, and concerns and maximize benefits and facilitators towards vaccination. 

Despite the compelling findings obtained, this study had limitations that merit discussion. Due to the emergence of the Omicron variant in December 2021, we changed the study design methodology, leading to potential selection bias in our sample. This bias may explain the higher booster acceptance than that reported by the PRDH. Our recruitment strategies resulted in a lower participation rate of men than women, with more participants having higher educational attainment. This selection bias is very common in questionnaires and studies conducted online [23]. Another possible limitation we considered in our study is the compensation ($30.00 gift card) given to participants upon completing the survey. Providing compensation in a study may introduce a bias in the sample, but it is difficult to know the direction of this bias. In our case, participants knew there was compensation, but they did not receive it until the end of the survey. The gift card was given without knowing the participant’s responses. Finally, some limitations are inherent to the HBM, including the exclusion of habitual behaviors (e.g., continuous preventive action) and access to information, which may influence the decision-making process. 

Notwithstanding the limitations of the study, our findings are important to understand COVID-19 vaccine booster acceptance in a Hispanic population with a high baseline vaccine approval. Our sample included adults living in Puerto Rico; to our knowledge, no studies have examined COVID-19 booster vaccination acceptance using the HBM in this sample. Moreover, our study was implemented during a critical period concerning the COVID-19 pandemic and the vaccine administration. This allowed an understanding of attitudes, beliefs, and susceptibility in Puerto Rico, where the baseline higher burden of chronic diseases and the aging of the population made COVID-19 booster acceptance increasingly important to decrease morbidity, quality of life, and mortality on the island. 

## 5. Conclusions

This study identified modifiable HBM predictors of COVID-19 booster dose hesitancy that may serve as targets for educational interventions or community-based strategies. Campaigns must be developed to help mitigate factors contributing to vaccine refusal. 

## Figures and Tables

**Table 1 vaccines-11-01426-t001:** Sociodemographic characteristics according to booster acceptance status (n = 784).

	Frequency (%)	Booster Acceptance Status		
Characteristic		Refusal * (n = 173)	Acceptance * (n = 611)	*p*-Value
**Age (years) ****				0.278
Mean (SD)	38.6 (14.1)	-------		
18–29 years	278 (35.7)	54 (19.4)	224 (80.6)	
30–45 years	259 (33.2)	69 (26.6)	190 (73.4)	
≥45 years *	240 (31.1)	48 (20.0)	192 (80.0)	
**Sex ****				0.663
Male	214 (27.8)	45 (21.0)	169 (79.0)	
Female	556 (72.2)	125 (22.5)	431 (77.5)	
**Marital status ****				0.053
Married, living with a partner	355 (45.6)	90 (25.4)	265 (74.6)	
Divorced, separated, widowed, single	424 (54.4)	83 (19.6)	341 (80.4)	
**Education level ****				0.255
High school diploma or less	57 (7.4)	15 (26.3)	42 (73.7)	
GED, some college, associate degree	159 (20.9)	40 (25.2)	119 (74.8)	
Bachelors, masters, and/or doctoral degree(s)	549 (71.7)	110 (20.0)	439 (80)	
**Annual income ****				0.007
≤$40,000	373 (54.5)	95 (25.5)	278 (74.5)	
>$40,000	312 (45.5)	53 (17.0)	259 (83.0)	
**Health insurance ****				0.049
No	30 (4.0)	11 (36.7)	19 (63.3)	
Yes	745 (96.0)	160 (21.5)	585 (78.5)	
**Importance of religion ****				0.070
Not important	194 (25.0)	32 (16.5)	162 (83.5)	
Somewhat important	184 (23.7)	42 (22.8)	142 (77.2)	
Important	398 (51.3)	99 (24.9)	299 (75.1)	

* Row percent presented in parenthesis. ** Age: 7 missing values. Sex: 14 missing values. Marital Status: 5 missing values. Education level: 19 missing values. Annual income: 99 missing values. Health insurance: 9 missing values. Importance of religion: 8 missing values.

**Table 2 vaccines-11-01426-t002:** Bivariate association between Health Belief Model constructs and booster acceptance status: PR-CEAL (n = 784).

	Booster Acceptance Status		
HBM Construct	Refusal * (n = 173)	Acceptance * (n = 611)	*p*-Value
Perceived susceptibility			
My chance of getting COVID-19 in the next few months is high. ^1^			0.348
Agree	81 (20.6)	312 (79.4)	
Disagree	91 (23.4)	298 (76.6)	
I am worried about the likelihood of getting COVID-19. ^2^			<0.01
Agree	107 (17.6)	501 (82.4)	
Disagree	65 (37.6)	108 (62.4)	
Getting COVID-19 is currently a possibility for me. ^3^			0.225
Agree	110 (20.7)	420 (79.3)	
Disagree	62 (24.6)	190 (75.4)	
Perceived severity			
The complications of contracting COVID-19 are serious. ^4^			<0.01
Agree	127 (18.1)	574 (81.9)	
Disagree	45 (55.6)	36 (44.4)	
I will be very sick if I get COVID-19. ^5^			0.236
Agree	55 (19.6)	225 (80.4)	
Disagree	117 (23.3)	385 (76.7)	
I am afraid of getting COVID-19. ^6^			<0.01
Agree	76 (16.0)	400 (84.0)	
Disagree	95 (31.1)	210 (68.9)	
Perceived benefits			
The booster is a good idea because it makes me feel less worried about catching COVID-19. ^7^			<0.01
Agree	68 (12.0)	500 (88.0)	
Disagree	105 (49.8)	106 (50.2)	
The booster will decrease my chances of getting COVID-19 or its complications. ^8^			<0.01
Agree	86 (14.0)	529 (86.0)	
Disagree	87 (52.4)	79 (47.6)	
Perceived barriers			
I worry the possible side effects of the booster would interfere with my usual activities. ^9^			<0.01
Agree	119 (35.4)	217 (64.6)	
Disagree	54 (12.1)	392 (87.9)	
I am concerned about the efficacy of the COVID-19 booster. ^10^			<0.01
Agree	121 (33.8)	237 (66.2)	
Disagree	52 (12.2)	373 (87.8)	
I am worried about the safety of the COVID-19 booster. ^11^			<0.01
Agree	118 (36.1)	209 (63.9)	
Disagree	55 (12.2)	397 (87.8)	
Cues to action			
I will take the COVID-19 booster only if I am given adequate information about it. ^12^			<0.01
Agree	82 (73.9)	29 (26.1)	
Disagree	64 (94.1)	4 (5.9)	
Already got the booster	27 (4.5)	577 (95.5)	
I will take the COVID-19 booster only if the vaccine is taken by many in the public. ^13^			<0.01
Agree	30 (69.8)	13 (20.2)	
Disagree	112 (84.2)	21 (15.8)	
Already got the booster	31 (5.1)	575 (94.9)	

* Row percent presented in parenthesis. ^1^ 2 missing values; ^2^ 3 missing values; ^3^ 2 missing values; ^4^ 2 missing values; ^5^ 2 missing values; ^6^ 3 missing values; ^7^ 5 missing values; ^8^ 3 missing values; ^9^ 2 missing values; ^10^ 2 missing values; ^11^ 5 missing values; ^12^ 1 missing values; ^13^ 2 missing values. HBM—health belief model.

**Table 3 vaccines-11-01426-t003:** Multivariable association between Health Belief Model constructs and booster refusal: PR-CEAL (n = 784).

HBM Construct	Crude PR (95% CI)	*p*-Value	Adjusted PR (95% CI)	*p*-Value
Perceived susceptibility				
My chance of getting COVID-19 in the next few months is high.				
Disagree	1.14 (0.84, 1.53)	0.407	1.06 (0.76, 1.47)	0.763
I am worried about getting COVID-19.				
Disagree	2.14 (1.57, 2.91)	<0.001	2.26 (1.61, 3.16)	<0.001
Getting COVID-19 is currently a possibility for me.				
Disagree	1.19 (0.87, 1.62)	0.284	1.17 (0.83, 1.64)	0.382
Perceived severity				
The complications of contracting COVID-19 are serious.				
Disagree	3.07 (2.18, 4.31)	<0.001	3.50 (2.41, 5.07)	<0.001
I will be very sick if I get COVID-19.				
Disagree	1.19 (0.86, 1.64)	0.296	1.27 (0.89, 1.81)	0.173
I am afraid of getting COVID-19.				
Disagree	1.95 (1.44, 2.64)	<0.001	2.01 (1.44, 2.80)	<0.001
Perceived benefits				
Getting the booster is a good idea because doing so will make me feel less worried about catching COVID-19.				
Disagree	4.58 (3.06, 5.64)	<0.001	4.20 (3.00, 5.90)	<0.001
Getting the booster will decrease my chances of getting COVID-19 or suffering from 1 or more of its complications.				
Disagree	3.75 (2.78, 5.05)	<0.001	3.70 (2.64, 5.18)	<0.001
Perceived barriers				
I worry that the possible side effects of the booster would interfere with my usual activities.				
Agree	2.92 (2.12, 4.04)	<0.001	2.47 (1.73, 3.51)	<0.001
I am concerned about the efficacy of the COVID-19 booster.				
Agree	2.76 (1.99, 3.82)	<0.001	2.50 (1.75, 3.58)	<0.001
I am worried about the safety of the COVID-19 booster.				
Agree	2.97 (2.15, 4.08)	<0.001	2.80 (1.96, 3.99)	<0.001
Cues to action				
I will take the COVID-19 booster only if I am given adequate information about it.				
Disagree	1.27 (0.92, 1.77)	0.146	1.26 (0.88, 1.81)	0.219
Already got the booster	0.061 (0.04, 0.09)	<0.001	0.062 (0.04, 0.10)	<0.001
I will take the COVID-19 booster only if the vaccine is taken by many in the public.				
Disagree	1.21 (0.81, 1.81)	0.360	1.30 (0.81, 2.09)	0.271
Already got the booster	0.073 (0.04, 0.12)	<0.001	0.08 (0.05, 0.15)	<0.001

PR—Prevalence Ratio. CI—Confidence Interval. HBM—Health Belief Model.

## Data Availability

The data presented in this study are available on request from the corresponding author. The data are not publicly available due to IRB’s privacy/ethical restrictions.

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
