# Peer review of "COVID-19 Vaccine Booster Hesitancy among Hispanic Adults: A Cross-Sectional Study from the Puerto Rico Community Engagement Alliance against COVID-19 Disparities (PR-CEAL)"

_vaccines, 2023, doi:10.3390/vaccines11091426_

Round 1
Reviewer 1 Report
In this paper Berrios et al. investigated the assess health belief correlates of COVID-19 vaccine booster uptake in Hispanic/Latino communities of Puerto Rico.
The manuscript is well written and a good survey has been designed. Also the results appear to be interesting, however, the experimental design was not completely appropriate.
As the Authors stated in the conclusions, the study design methodology changed during the study from a face-to-face to an online survey, leading to a selection bias. Also, both the face-to-face (supermarket) and the online survey led to a higher recruitment of women. (more than 70%).
The manuscript require minor revisions:
1. Because of a higher number of women recruited, the statistical analysis should be made also taken in consideration only the female population.
2. Only the number of partecipants included is reported. How many people has partecipated to the survey? How many face to face and how many on line? What was the ratio men/ women? How many people has completed the survey? 787 is the number of partecipants included or is the number of survey completed? If the number is different, what were the inclusion/exclusion criteria of the survey selection for the statistical analysis?
English should be improved
Reviewer 2 Report
Abstract:
The abstract lacks specific details regarding the sampling method used to recruit the 787 participants, which could affect the representativeness of the sample. Furthermore, the abstract does not provide information about the demographic characteristics of the participants, making it difficult to assess the generalizability of the findings.
Introduction:
The introduction lacks clarity and structure, making it difficult to follow the flow of information. The timeframe of the study is not clearly stated, as the introduction begins with the reported cases and deaths up until February 2023 but later refers to education strategies in December 2020. The introduction also lacks a clear research gap or objective statement, making it challenging to understand the purpose of the study. Additionally, while references are provided to support statements, the introduction does not critically evaluate or discuss the existing literature in a coherent manner. A more focused and well-organized introduction would enhance the overall quality of the manuscript.
Methodology:
It does not provide a clear explanation of the sampling method or how the sample size was determined. The recruitment process is mentioned briefly but lacks specific details regarding the selection criteria for visiting supermarkets and community settings. The shift to online recruitment due to the Omicron variant is mentioned but not justified or discussed further. The survey instrument is not provided, making it difficult to assess the validity and reliability of the measurements used. The statistical analysis section lacks information on the specific statistical tests used, the criteria for selecting variables in the adjusted models, and any potential covariates that were considered.
Results:
The demographic characteristics of the participants are summarized, but there is no analysis of the demographic factors' association with booster refusal. The statement that "almost all the HBM constructs were associated with booster refusal" is vague and does not provide specific details or effect sizes. The associations between specific HBM constructs and booster refusal are mentioned briefly but lack in-depth analysis. The presentation of adjusted prevalence ratios lacks clarity, and there is no mention of other potential covariates that were considered in the analysis.
Discussion:
The discussion lacks a thorough analysis and interpretation of the findings. The statement that booster acceptance was high in the sample is not adequately supported or contextualized. The discussion fails to provide a clear explanation of the barriers, misinformation, and beliefs that contribute to booster refusal. The limitations of the study are briefly mentioned but not thoroughly discussed in terms of their impact on the findings. The recommendations provided lack specificity and do not offer practical solutions based on the study's results.
The conclusion merely restates the findings without offering further insights or implications.
Reviewer 3 Report
The manuscript (ID: vaccines-2457587) aimed to assess health belief correlates of COVID-19 vaccine booster uptake among the Hispanic adults in Puerto Rico.
However, there are numerous omissions and errors in the paper (the Methods, Results and Discussion sections):
- Lines 15-16: As part of this sentence of `Abstract`, specify the study design applied in this paper.
- Lines 16-17: Align with the claims presented in the text on Lines 94-102. Explain whether the recruitment of participants for this work was `via an online` (Line 17) or `in-person participant recruitment started on December 30, 2021, and consisted of visiting supermarkets and community settings` as stated on Lines 94- 95?
- Line 41: Add a new paragraph in which it should be stated when the administration of booster doses began in Puerto Rico, what indications were foreseen for the booster dose, whether the vaccine was paid for or free. Present data on adverse reactions following COVID-19 primary vaccination in Puerto Rico.
- Lines 77-106: Completely reorganize the text, as follows:
- Specify the study design applied in this paper. Nowhere in the paper is the design of the study indicated.
- Enter new subsections
- `Study population`,
- `Study sample`, determine `Response rate`,
- `Questionnaires`,
- `Data curation`.
- Line 154: The numbers/results in Table 1 are questionable in their entirety. Check this.
- Line 162: The numbers/results in Table 2 are questionable in their entirety. Check this.
- Lines 177-195: Very sparse Discussion section. A comprehensive comparison with the results of similar research in the world is lacking. The literature on this topic is numerous and available.
- Lines 197-209: Some limitations of this work are stated. Many of these limitations could have been eliminated either during the planning and/or conducting of the study, or during data analysis in this manuscript.
The quality of English language is appropriate.
Round 2
Reviewer 2 Report
Dear Author, The manuscript is now in better form, but few changes are required.
1- Please organize the tables (particularly Table 2 and 3) to enhance the readability. Is it possible to add some graphs of these tables???
Otherwise, just arrange them, as so long and complex tables reduce the readability.
2- Discussion is very concise, it would be better to add some more citations and discussion in this section.
3- There must be a list/ table of abbreviations at the end of the manuscript.
4- There should be proof reading of the manuscript. as few typos and long sentences are found. Please read and correct in the whole manuscript.
Reviewer 3 Report
Thank you for the opportunity to re-review the manuscript (ID: vaccines-2457587). The authors responded to most of my comments, but not all. I thank the authors.
However, it is worth noting again my comment in the previous review, which reads:
- Line 154 (now in revised version at Line: 170): The numbers/results in Table 1 are questionable in their entirety. Check this.
- Line 162 (now in the revised manuscript at Line: 180): The numbers/results in Table 2 are questionable in their entirety. Check this.
Namely:
t The authors stated in their response that a reanalysis produced the same results.
o But, my comments were about `numbers/results`. That means both numbers and results. It is known that the primary question was numbers, and the secondary question was the results that depend on them.
o But the numbers in the Tables remained the same in the original version and in the revised manuscript.
o It is necessary to check the numbers in Tables 1 and 2 and correct them, I am citing only 1 example: for the variable `30-45 years`, did the number `259 (33.2)` represent the sum of the numbers `48 (28.2)` and `122 (71.8)`?
o In this way, check the variables `45 + years`, `Health insurance, No/Yes`.
o Then check for the variable `Sex`, are the numbers for `Refusal` and `Acceptance` correctly represented?
t In order to improve the clarity and transparency of the results, do the following:
- On Lines 167-169, enter figures (78%, number/total number; 22%, number/total number), it is not a good practice to enter only % for this important variable;
- In the titles and subheaders of Tables 1 and 2, state the numbers for the total number of respondents, that is, the numbers for `Refusal` and `Acceptance`;
- According to `footer to the table to set the missing values`, which was added for Table 1 in the revised manuscript, enter the data in a similar way under Table 2.
t In the limitations of this paper, discuss whether the `$30 gift card` influenced the results of this research.
The quality of English language is appropriate.
Round 3
Reviewer 3 Report
The authors addressed all my comments and provided appropriate explanations. All necessary corrections have been made in the revised version of this manuscript. Thanks to the authors.
The quality of English language is appropriate.
Author Response
Thanks